# Enhancing the Performance of Vibration Energy Harvesting Based on 2:1:2 Internal Resonance in Magnetically Coupled Oscillators

**DOI:** 10.3390/mi16010023

**Published:** 2024-12-27

**Authors:** Shakiba Dowlati, Najib Kacem, Noureddine Bouhaddi

**Affiliations:** Department of Applied Mechanics, FEMTO-ST Institute, CNRS, University of Franche-Comté, F-25000 Besançon, France; shakiba.dolati@gmail.com (S.D.); noureddine.bouhaddi@univ-fcomte.fr (N.B.)

**Keywords:** vibration energy harvesting, magnetically coupled oscillators, internal resonances, multi-objective optimization

## Abstract

An electromagnetic vibration energy harvester with a 2:1:2 internal resonance (IR) is proposed, allowing for the simultaneous activation of two IRs within the system in order to enhance its performance in terms of bandwidth and harvested power. The device consists of three magnetically coupled oscillators separated by an adjustable gap to tune the system eigenfrequencies and achieve a 2:1:2 IR. Numerical investigations are conducted to predict the behavior of the proposed device, and a multi-objective optimization procedure is employed to enhance the harvester’s performance by introducing mass perturbations. The experimental validation of the optimized design is performed while highlighting the benefits of internal resonance, and the obtained results are in good agreement with the theoretical findings. The results indicate that incorporating two internal resonances into the harvester enhances its performance compared to the harvesters reported in the literature. The harvester achieves an ${\mathrm{SFoM}}_{\mathrm{BW}}$ of 7600 kg/m^3^, reflecting a high average power density over a broad bandwidth.

## 1. Introduction

Modern electronic devices are becoming increasingly smaller, wireless, and have lower power consumption, enabling longer-term functionality. Conventional electrochemical batteries used to power these devices face several limitations, such as the need for regular charging and replacement [1]. The high costs and environmental impacts of battery disposal further complicate this problem [2]. Energy-harvesting technologies offer a promising alternative by converting ambient energy sources into electricity [3]. Vibration energy has attracted great attention due to its ubiquity and abundance in our surroundings. Among the different transduction mechanisms, electromagnetic transduction, with a long lifespan and relatively high current output, is a well-established technique [2].

Ambient vibrations exhibit broadband, multi-frequency characteristics. The main challenge in vibration energy harvesting (VEH) is the frequency mismatch between the natural frequency of the harvester and the ambient vibrations, which considerably reduces the efficiency of the harvester [4]. Various techniques have been developed to address this issue and enhance the harvester’s performance. One technique involves tuning the harvester’s resonance frequency to align with the dominant vibration frequency. This can be achieved with mechanical methods, such as adjusting geometric parameters, system stiffness, and applying an axial preload [5,6], or electrical methods, such as manipulating circuit parameters, adding shunt circuits, or using impedance-matching networks [7,8]. Li et al. [9] reported a tunable VEH device that adjusts its resonant frequency by altering the spring length. Morel et al. [8] demonstrated the impact of electrically induced damping and stiffness on the frequency response for a piezoelectric harvester. Multimodal techniques are another approach that can be implemented through a multi-frequency array harvester [10]. Sari et al. [11] proposed an electromagnetic harvester with an array of 40 cantilevers that generate steady power over a uniform spectrum of natural frequencies. Yu et al. [12] proposed a multimodal piezoelectric harvester with hinged supports and rotating shafts that target the first and third resonant frequencies. Zergoune et al. [13] studied energy localization in a periodic multimodal harvester using mass mistuning to improve the harvested power density.

Linear harvesters have intrinsic limitations that restrict the potential of these techniques. Incorporating nonlinearities into the harvester design can improve the performance of the device in terms of the frequency bandwidth and output power. Monostable nonlinear VEH has been achieved using stretching strain in clamped resonators, nonlinear magnetic levitation, and structures with magnetic masses [14,15]. Abed et al. [16] developed a multimodal VEH using arrays of coupled levitated magnets to broaden the frequency bandwidth by leveraging nonlinear coupling among three coupled magnets. Kankana et al. [17] designed a nonlinear wideband harvester with a tapered spring architecture, displaying a nonlinear restoring force. Several studies have also developed bistable harvesters incorporating repulsive magnetic forces [18,19,20]. Multi-DOF bistable harvesters with magnetic coupling have been developed to enhance performance in broader frequency bands [21,22,23,24,25]. Podder et al. [26] proposed a wideband nonlinear VEH that can switch between tunable bistable-quadratic, monostable-quartic, and bistable-quartic potentials using induced dynamical nonlinearities.

While nonlinear harvesting devices can enlarge the frequency bandwidth, these extensions tend to occur in either higher or lower frequencies. The internal resonance phenomenon in nonlinear systems causes the amplitude-frequency response curves to exhibit bending toward both lower and higher frequencies. This double-bending feature can significantly increase frequency bandwidth [27]. This phenomenon arises from nonlinear intermodal interactions when linear natural frequencies of the system are commensurate or nearly commensurate [28,29]. Nonlinear systems have intermodal coupling where various modes can be activated independently by varying the external excitation [30]. Intermodal coupling, necessary for IR, results from the nonlinearities in the system. Nonlinearity can be introduced in several ways, such as geometric design and magnetic forces. Magnetic forces, however, offer the additional advantage of simultaneously tuning the natural frequencies of the system and introducing nonlinearities [31]. Xie et al. [32] proposed a magnetically coupled T-shaped VEH with magnetic couplings and axial loading based on a 1:2:3:4 IR. Aouali et al. [31] investigated the 2:1 IR in a hybrid nonlinear VEH experimentally. Garg and Dwivedy [33], as well as Aravindan and Ali [34], analytically studied cantilever piezoelectric VEHs with a 1:3 IR. Jiang et al. [35] investigated a piezoelectric VEH subjected to axial loading through analytical and numerical methods. The amplitude-frequency response curves exhibited dual jumps, bending to the left and right, indicating a hardening nonlinearity that contributed to broadband harvesting. The L-shaped cantilever-based beam structure is another approach to achieve internal resonances by combining two beams at an angle with smaller support structures and mass [36,37]. Bao et al. [38] studied a pendulum-based VEH comprising a piezoelectric cantilever with a magnetic pendulum and base magnet setup. The IR arose from the nonlinear coupling between the two-dimensional pendulum motion and the beam-bending vibration. Sun et al. [39] proposed a double pendulum-based VEH to overcome the limitations of ultra-low natural frequencies, using the second mode of a double pendulum along with a piezoelectric cantilevered beam, achieving an IR ratio of 2:1:2. Yang and Towfighian [40] reported a hybrid nonlinear VEH combining the concepts of bistability and IR. Despite extensive studies on modal interactions in nonlinear VEH, to the best of the authors’ knowledge, the exploration of tuning multiple internal resonances to enhance harvester performance has not yet been investigated.

In this paper, an electromagnetic VEH with a 2:1:2 IR is proposed to activate two IRs simultaneously, which results in broadband energy harvesting. The proposed device consists of three coupled magnets mounted at the center of a spiral-shaped spring and wrapped with copper coils. The behavior of the proposed device is studied numerically. A multi-objective optimization procedure is performed to maximize the frequency bandwidth and harvesting efficiency of the VEH. This is achieved through numerical studies involving the introduction of mass perturbation into the oscillators. Experimental investigations are then conducted to study the performance in terms of the harvested power and bandwidth. The experimental results, obtained under harmonic excitation, validate the presence of two IRs and show good agreement with the theoretical findings.

## 2. Design and Modeling of 2:1:2 IR-Based VEH

An equivalent spring-mass model of the proposed harvester is illustrated in Figure 1a. The harvester consists of three coupled spiral springs with magnets as proof masses in the center. For electromagnetic transduction, the two identical magnets ($m_{1}$ and $m_{3}$) are wrapped with coils. The parameters $c_{1}$ and $c_{2}$ represent the total damping coefficients, which are the sums of the mechanical damping coefficient $c_{m}=2\xi_{m}m\omega_{0}$ and the electrical damping $c_{e}$. Therefore, the subsystem without a coil exhibits only mechanical damping, while the two subsystems wrapped with coils experience the combined effects of both electrical and mechanical damping. $d_{1}$ denotes the initial distance between the first and second magnets, and $d_{2}$ represents the initial distance between the second and third magnets. $k_{mg,1}$ and $k_{mg,2}$ are the nonlinear coupling stiffnesses between the two repulsive magnets. $x_{1}$, $x_{2}$, and $x_{3}$ are the relative displacements of the corresponding masses. The springs are magnetically coupled with nonlinear repulsive forces. The distinct stiffnesses of the springs and the coupling through repulsive magnetic forces allow for tuning the natural frequencies. In addition to tuning the natural frequency ratio through the magnets’ gap distance, magnetic coupling also introduces quadratic nonlinearities into the system. The schematic of a single-subsystem electromagnetic VEH, consisting of a spring and a magnet wrapped with a coil, is shown in Figure 1d.

The governing equations of the nonlinear electromagnetic VEH device under harmonic base excitation can be expressed as
(1)$$\begin{array}{c} m_{n}\ddot{x_{n}}+c_{n}\dot{x_{n}}+k_{n}x_{n}+F_{mg}^{n}=-m_{n}{\ddot{X}}_{g}, \end{array}$$
(2)$$\begin{array}{c} i_{n}\left(t\right)=\frac{\delta_{em}}{R_{load}+R_{int}}\;{\dot{x}}_{n},\;\;n=1,2,3. \end{array}$$
where $m_{n}$ represents the equivalent masses of the magnets as the proof masses of the springs with linear mechanical stiffnesses $k_{n}$. $c_{n}$ is the corresponding damping coefficient. The relative displacement of each magnet is denoted as $x_{n}$. $\delta_{em}$ is the electromechanical coupling coefficient. $R_{load}$ and $R_{int}$ are the load resistance and internal resistance of the identical coils, respectively. ${\ddot{X}}_{g}=X_{g}\sin\left(\Omega t\right)$ is the acceleration of the harmonic basis excitation, where $X_{g}$ is the base excitation amplitude and Ω is the excitation frequency.

The steady-state response of the harmonic excitation is given by $x_{n}=X_{n}\sin\left(\omega t-\phi \right)$. The electrical power generated at each coil is $P_{n}\left(t\right)=c_{e}{\dot{x}}_{n}^{2}$, where $c_{e}=\frac{\delta_{em}^{2}}{\left(R_{load}+R_{int}\right)}$ is the electrical damping [41]. The instantaneous harvested power, which represents the power dissipated by the resistance in each energy-harvesting circuit corresponding to each subsystem, can be expressed as follows:(3)$$\begin{array}{c} P_{n}\left(t\right)=R_{load}i_{n}^{2}\left(t\right). \end{array}$$

The average load power harvested across the load resistances over an oscillation cycle from *t* to t+T is
(4)$$\begin{array}{c} P_{load}=\sum_{n=1}^{N}R_{load}{\left(\frac{\omega_{0}\delta_{em}}{\left(R_{load}+R_{int}\right)}\right)}^{2}{\left|X_{n}\right|}^{2}. \end{array}$$

The nonlinear magnetic force $F_{mg}^{n}$ can be written as [42]
(5)$$F_{mg}^{n}=\frac{\mu_{0}Q_{M_{n}}}{4\pi }\left(\frac{Q_{M_{n-1}}}{{\left(d_{n-1}+x_{n-1}-x_{n}\right)}^{2}}-\frac{Q_{M_{n+1}}}{{\left(d_{n}+x_{n}-x_{n+1}\right)}^{2}}\right),\;n=1,2,3,$$
where $d_{n}$ is the initial gap between the magnets $Q_{M_{n}}$ and $Q_{M_{n+1}}$, $Q_{M_{n-1}}=Q_{M_{n}}=Q_{M_{n+1}}=Q_{M}$ is the magnetization moment of the identical magnets, and $\mu_{0}=4\pi \times 10^{-7}$ H.m^−1^ refers to the permeability of free space. By approximating the nonlinear magnetic forces using a second-order Taylor series and neglecting higher-order terms [31], Equation (Equation 1) under harmonic base excitation can be rewritten as
(6)$$\begin{array}{cc} \ddot{x_{1}}+\mu_{1}\dot{x_{1}}+{\tilde{\omega }}_{1}^{2}\left[\left(1+\beta_{1}\right)x_{1}-\beta_{1}x_{2}\right]-f_{mg,nl}^{1}{\left(x_{1}-x_{2}\right)}^{2} & =-X_{g}\sin\left(\Omega t\right)\\ \ddot{x_{2}}+\mu_{2}\dot{x_{2}}+{\tilde{\omega }}_{2}^{2}\left[\left(1+\beta_{2}+\beta_{3}\right)x_{2}-\beta_{2}x_{1}-\beta_{3}x_{3}\right] \end{array}$$
(7)$$\begin{array}{cc} \;\;\;\;\;\;\;+f_{mg,nl}^{2}{\left(x_{2}-x_{1}\right)}^{2}-f_{mg,nl}^{3}{\left(x_{2}-x_{3}\right)}^{2} & =-X_{g}\sin\left(\Omega t\right) \end{array}$$
(8)$$\begin{array}{cc} \ddot{x_{3}}+\mu_{3}\dot{x_{3}}+{\tilde{\omega }}_{3}^{2}\left[\left(1+\beta_{4}\right)x_{3}-\beta_{4}x_{2}\right]+f_{mg,nl}^{4}{\left(x_{3}-x_{2}\right)}^{2} & =-X_{g}\sin\left(\Omega t\right) \end{array}$$
where $\mu_{1}$, $\mu_{2}$, and $\mu_{3}$ are the normalized damping coefficients, representing the combined effect of the mechanical and electrical damping coefficients, and ${\tilde{\omega }}_{n}$ is the natural frequency of the uncoupled oscillators. $\beta_{1}$, $\beta_{2}$, $\beta_{3}$, and $\beta_{4}$ are the linear coupling coefficients, where $k_{mg,l}^{1}$ and $k_{mg,l}^{2}$ are the linear coupling stiffnesses between the successive magnets, respectively. $k_{mg,nl}^{1}$ and $k_{mg,nl}^{2}$ are the quadratic nonlinear coupling stiffnesses between the two successive magnets, respectively. $f_{mg,nl}^{1}$, $f_{mg,nl}^{2}$, $f_{mg,nl}^{3}$, and $f_{mg,nl}^{4}$ are the normalized nonlinear stiffnesses. These parameters are defined in the following equations:(9)$$\begin{array}{cc} \mu_{1} & =2\xi_{m}\omega_{0}+\frac{\delta_{em}^{2}}{m_{1}\left(R_{load}+R_{int}\right)},\;\mu_{2}=2\xi_{m}\omega_{0},\\ \mu_{3} & =2\xi_{m}m_{1}\omega_{0}+\frac{\delta_{em}^{2}}{m_{1}\left(R_{load}+R_{int}\right)},\;{\tilde{\omega }}_{n}=\sqrt{\frac{k_{n}}{m_{n}}},\\ \beta_{1} & =\frac{k_{mg,l}^{1}}{k_{1}},\;\beta_{2}=\frac{k_{mg,l}^{1}}{k_{2}},\;\beta_{3}=\frac{k_{mg,l}^{2}}{k_{2}},\;\beta_{4}=\frac{k_{mg,l}^{2}}{k_{1}},\\ k_{mg,l}^{1} & =\frac{Q_{M}^{2}\mu_{0}}{2\pi d_{1}^{3}},\;k_{mg,l}^{2}=\frac{Q_{M}^{2}\mu_{0}}{2\pi d_{2}^{3}},\\ k_{mg,nl}^{1} & =\frac{3Q_{M}^{2}\mu_{0}}{4\pi d_{1}^{4}},\;k_{mg,nl}^{2}=\frac{3Q_{M}^{2}\mu_{0}}{4\pi d_{2}^{4}},\\ f_{mg,nl}^{1} & =\frac{k_{mg,nl}^{1}}{m_{1}},\;f_{mg,nl}^{2}=\frac{k_{mg,nl}^{1}}{m_{2}},\;f_{mg,nl}^{3}=\frac{k_{mg,nl}^{2}}{m_{2}},\;f_{mg,nl}^{4}=\frac{k_{mg,nl}^{2}}{m_{1}}. \end{array}$$

The three natural frequencies of the system can be obtained by solving the eigenvalue problem (K−λM)ϕ=0, where M is the unit mass matrix and K is the linear stiffness matrix, given by
(10)$$\begin{array}{c} K=\left[\begin{array}{ccc} {\tilde{\omega }}_{1}^{2}\left(1+\beta_{1}\right) & -{\tilde{\omega }}_{1}^{2}\beta_{1} & 0\\ -{\tilde{\omega }}_{2}^{2}\beta_{2} & {\tilde{\omega }}_{2}^{2}\left(1+\beta_{2}+\beta_{3}\right) & -{\tilde{\omega }}_{2}^{2}\beta_{3}\\ 0 & -{\tilde{\omega }}_{3}^{2}\beta_{4} & {\tilde{\omega }}_{3}^{2}\left(1+\beta_{4}\right) \end{array}\right] \end{array}$$The solutions are distinct real eigenvalues $\lambda_{n}=\omega_{n}^{2}$. Since the three oscillators are coupled through linear and quadratic stiffnesses, one can exploit modal interactions [43] and trigger two IRs by adjusting the harvester design to obtain the desired commensurability of the natural frequencies $2\omega_{1}=\omega_{2}=2\omega_{3}$.

The proposed harvester is fabricated, as shown in Figure 1b, using the spring illustrated in Figure 1c. The spring consists of two spiral-shaped cantilevers that support a vertically movable central stage. The spring arms are 0.6 mm in width and 0.06 mm in thickness, with 3.5 turns. This spiral design allows for more length in a compact area, thus lowering the natural frequency of the spring. A NdFeB magnet is bonded to the circular stage of 5 mm diameter, suspended by the two spring arms.

The energy is harvested from the first and third oscillators (subsystems 1 and 3). Therefore, coils couple these two subsystems to load resistances, and the generated voltage is monitored. The mechanical damping coefficient is proportional to the 3 dB bandwidth of the resonant peak relative to the peak frequency [44]. An open-loop circuit is used to measure the vibration of the spring using a laser Doppler vibrometer. The mechanical damping coefficient $\xi_{m}$ is experimentally identified as 0.16%. The electromagnetic coupling coefficient is $\delta_{em}=0.14$ V.S/m. The design parameters of the proposed harvester are listed in Table 1. The linear stiffness coefficient is determined using FEM in Ansys 2022 R1 by simulating the force–deflection relationship and fitting the results to estimate the stiffness. The mechanical damping coefficient $\xi_{m}$ is experimentally identified using the half-power bandwidth method (−3 dB method), calculated as $\xi_{m}=\Delta f/2f_{0}$, where Δf is the 3 dB bandwidth and $f_{0}$ is the resonant frequency. The electromechanical coupling coefficient $\delta_{em}$ represents the relationship between mechanical motion and induced voltage in the coil, given by
$$\delta_{em}=\frac{d\phi }{dx}=2N\pi R\int \frac{B(x,h_{mag},R)}{dx}dh,$$
where *R* is the coil radius, *B* is the magnetic field, and $h_{mag}$ is the height of the magnet. This coefficient is determined using the FEMM 4.2 software for magnetic field distribution, influencing the electrical output of the system.

Figure 2 illustrates that by adjusting the gap between the successive magnets, the ratios between the system’s natural frequencies can be tuned. At specific distances of $d_{1}^{*}=d_{2}^{*}=9.6$ mm, the ratios reach a value of $2.02\omega_{1}=\omega_{2}=1.99\omega_{3}$.

## 3. Solving and Optimization Procedures

### 3.1. Numerical Solution

The Finite Difference Method (FDM) is a numerical technique used to find approximate solutions to linear and nonlinear ordinary differential equations [45,46]. It involves approximating the differential operator by replacing the derivatives with a finite difference quotient. For the implementation of the FDM on the equations of motion for the proposed VEH, a Taylor-series expansion is used to extract the fourth-order centered difference scheme given by
(11)$$\begin{array}{cc} x_{n}^{\prime } & :\frac{-x_{n}^{i+2}+8x_{n}^{i+1}-8x_{n}^{i+1}+x_{n}^{i+2}}{12h} \end{array}$$
(12)$$\begin{array}{cc} \;\;\;\;x_{n}^{\prime \prime } & :\frac{-x_{n}^{i+2}+16x_{n}^{i+1}-30x_{n}^{i}+16x_{n}^{i+1}-x_{n}^{i+2}}{12h^{2}} \end{array}$$
where *h* must be small enough to provide a good approximation of the derivative of the quotient. Substituting Equations (Equation 11) and (Equation 12) into Equations (Equation 6)–(Equation 8) results in a finite set of coupled nonlinear algebraic equations, which is solved using the arclength continuation method [47] with respect to the drive frequency to capture the frequency response of the system, as shown in Figure 3.

Using the parameters of the energy harvester provided in Table 1, the harvester exhibits a 2:1:2 IR with $\omega_{1}=50.7$ Hz, $\omega_{2}=102.4$ Hz, and $\omega_{3}=51.45$ Hz. Figure 3 illustrates the frequency response of the harvested power of the two subsystems at an acceleration of 0.4 g. The response exhibits the double-jumping phenomenon, characterized by two peaks bending toward higher and lower frequencies. This indicates an exchange of energy between the three modes. This phenomenon can contribute to broadband vibration energy harvesting, where the resonance range defines the effective bandwidth. However, the power peak and the response branches are not symmetric, adversely affecting the harvester’s efficiency. Adjusting the ratio of natural frequencies can maintain the symmetry in the frequency responses or shift the response to higher and lower frequencies. This can be performed by introducing perturbations into the system to tune the bifurcation topology of the proposed VEH and improve its performance. This opens the way to investigate the optimal perturbation strategy that maximizes the frequency bandwidth and the harvested power.

### 3.2. Multi-Objective Optimization

When designing an energy harvester with a 2:1:2 internal resonance, the accurate ratio of frequencies that results in a symmetric response must be determined. By shifting the frequency responses in opposite directions from the center frequency, the resonance range can be broadened, leading to a significant increase in the harvester’s bandwidth. Introducing mass perturbations into the two subsystems can create the desired ratio of natural frequencies. The optimal amount of these mass perturbations has to be determined through an optimization procedure. Therefore, a multi-objective optimization procedure is performed to achieve a wider bandwidth and higher power through mass perturbations.

Multi-objective optimization is performed on the system of equations of motion as follows: (13)$$\begin{array}{cc} \ddot{x_{1}}+\mu_{1}\dot{x_{1}}+\alpha_{1}^{-1}{\tilde{\omega }}_{1}^{2}\left[\left(1+\beta_{1}\right)x_{1}-\beta_{1}x_{2}\right]-\alpha_{1}^{-1}f_{mg,nl}^{1}{\left(x_{1}-x_{2}\right)}^{2} & =-X_{g}\sin\left(\Omega t\right)\\ \ddot{x_{2}}+\mu_{2}\dot{x_{2}}+{\tilde{\omega }}_{2}^{2}\left[\left(1+\beta_{2}+\beta_{3}\right)x_{2}-\beta_{2}x_{1}-\beta_{3}x_{3}\right] \end{array}$$(14)$$\begin{array}{cc} \;\;\;\;\;\;\;\;\;\;+f_{mg,nl}^{2}{\left(x_{2}-x_{1}\right)}^{2}-f_{mg,nl}^{3}{\left(x_{2}-x_{3}\right)}^{2} & =-X_{g}\sin\left(\Omega t\right) \end{array}$$(15)$$\begin{array}{cc} \ddot{x_{3}}+\mu_{3}\dot{x_{3}}+\alpha_{2}^{-1}{\tilde{\omega }}_{3}^{2}\left[\left(1+\beta_{4}\right)x_{3}-\beta_{4}x_{2}\right]+\alpha_{2}^{-1}f_{mg,nl}^{4}{\left(x_{3}-x_{2}\right)}^{2} & =-X_{g}\sin\left(\Omega t\right) \end{array}$$
where $\alpha_{1}$ and $\alpha_{2}$ are the mass perturbations applied to subsystems 1 and 3, respectively. The objectives of the optimization problem are to maximize the total bandwidth and the power harvested from these two subsystems. The optimization problem can be formulated as
(16)$$\begin{array}{cc} f_{1}\left(\alpha_{1},\alpha_{2}\right) & =\sum_{n}\left(B_{n}\right),\;\;\;\;\;\;\;\;\;\;\;\;\;\;\;\;\;\; \end{array}$$
(17)$$\begin{array}{cc} f_{2}\left(\alpha_{1},\alpha_{2}\right) & =\sum_{n}R_{load}^{2}{\left(\frac{\delta_{em}}{\left(R_{laod}+R_{int}\right)}\right)}^{2}(\omega_{n}^{2}|x_{n1}^{max}{|}^{2}+\omega_{n}^{2}x_{n2}^{max}{|}^{2})\;n=1,3. \end{array}$$
where $f_{1}\left(\alpha_{1},\alpha_{2}\right)$ is the objective function for maximizing the total bandwidth and $B_{n}$ is the bandwidth of the $n^{th}$ subsystem. The half-power bandwidth is defined within each branch considering the bandwidth around each peak, rather than the total range between peaks. $f_{2}\left(\alpha_{1},\alpha_{2}\right)$ is the function to maximize the power harvested from subsystems 1 and 3. $x_{n1}^{max}$ and $x_{n2}^{max}$ are the two peaks of power that occur due to the presence of two peaks in the IR frequency response for each subsystem.

With two objective functions, the problem is called a multi-objective optimization problem (MOOP), formulated as
(18)$$\begin{array}{c} \left\{\begin{array}{c} \begin{array}{ccc} \min_{\alpha_{1},\alpha_{2}} & \; & F\left(f_{1}\left(\alpha_{1},\alpha_{2}\right),f_{2}\left(\alpha_{1},\alpha_{2}\right)\right),\\ \mathrm{subject}\;\mathrm{to} & \; & \left\{\begin{array}{c} \begin{array}{cc} 0.95\times m_{1}\leq & \alpha_{1}\leq m_{1},\\ m_{1}\leq & \alpha_{2}\leq 1.05\times m_{1}. \end{array} \end{array}\right. \end{array} \end{array}\right. \end{array}$$

NSGA-II, an extension of the Non-dominated Sorting Genetic Algorithm (NSGA), is an effective tool for handling MOOPs [48,49]. The MATLAB R2023a function gamultiobj, which implements a variation of the Non-Sorting Genetic Algorithm (NSGA-II), is used in this work. Figure 4 shows the Pareto front to maximize the harvested power and the frequency bandwidth simultaneously.

The optimization procedure identified a set of non-dominated solutions where both power and bandwidth are simultaneously optimized. The best solution on the Pareto front for maximizing both objective functions is chosen by prioritizing the bandwidth, and the optimal set of mass perturbations is listed in Table 2. These mass perturbations are introduced into the system, and the frequency responses of the energy harvested from the two subsystems are obtained and illustrated in Figure 5. The results show that the optimized response exhibits nearly symmetrical power peaks for the two subsystems.

## 4. Experimental Results and Discussion

An electrodynamic shaker was used to generate predefined vibration characteristics. Electric currents flowing through a coil within the shaker’s magnetic field generated mechanical vibrations. The drive current and magnetic field characterizations of the shaker determine the force generated for acceleration. This drive signal was generated by a computer-controlled signal generator and subsequently amplified by a power amplifier. A feedback control loop is required to ensure the stability and accuracy of the vibration acceleration and frequency. This feedback loop was established by attaching an accelerometer to the shaker platform. The excitation signal was produced using the signal generator in response to the accelerometer’s feedback. The output data, such as voltages, amplitudes, and accelerations, were recorded by an “m + p VibPilot” computer monitor connected to a computer for storage. Preliminary experiments were carried out to tune the desired ratio of the natural frequencies using the experimental setup shown in Figure 6. The harvester in Figure 1b was mounted on the shaker and subjected to base harmonic excitation produced by a shaking table and its amplifier. Feedback from an accelerometer mounted on the base enabled constant base acceleration levels to be maintained. As the magnetic masses underwent a relative motion toward the coils, current flowed through the load resistance, and the voltage across them was measured. The materials and geometric parameters of the magnet and the coil are listed in Table 3.

The proposed design had a 2:1:2 commensurability ratio between the frequencies of its first three modes of vibration, experimentally determined to be $\omega_{1}=50.7$ Hz, $\omega_{2}=102.4$ Hz, and $\omega_{3}=51.25$ Hz. This configuration led to the activation of two IRs due to the quadratic coupling responsible for multimodal interactions. The voltage-frequency responses of both subsystems were investigated under identical load resistances of 25 Ω.

Figure 7a shows the power-frequency responses of subsystems 1 and 3 before optimization. The responses were obtained under an external harmonic base excitation of 0.4 g, with a forward and backward frequency sweep around the first primary resonance. Although the frequency responses show an M-shaped internal resonance behavior, the two peaks are not symmetrical. This asymmetry prevents consistent power across the bandwidth and limits the maximum harvested power.

The objective was to maximize the harvested power and bandwidth simultaneously. Mass perturbations were introduced into subsystems 1 and 3 using the values in Table 2, and the resulting frequency response is shown in Figure 7b. The results illustrate two IRs that shifted away from the center frequency. The power peaks are nearly symmetrical, maximizing the bandwidth and power. The bandwidth of each subsystem is around 1.9 Hz. The total bandwidth ranges from 49.8 to 52.4 Hz, with a total operational bandwidth of 2.6 Hz. Figure 8 shows that the numerical and experimental frequency responses are globally in good agreement. Future studies could be conducted to calibrate this model so that it becomes a useful predictive tool during the design phase of new VEH devices using multiple internal resonances.

Figure 9a shows the frequency response of the harvested power at an acceleration of 0.6 g. It can be observed that increasing the excitation causes the distance between the two peaks to become larger. The frequency bandwidth becomes broad, and the peak power also increases. The bandwidth of subsystems 1 and 3 is almost 2.1 Hz. The total bandwidth ranges from 49.6 to 52.7 Hz, representing an effective frequency bandwidth of 3.1 Hz.

Figure 9b displays the power frequency response of the harvester at an acceleration of 0.8 g. The frequency bandwidth becomes broad, and the peak power also increases. The bandwidth of subsystems 1 and 3 is almost 2.3 Hz. The total frequency bandwidth ranges from 49.4 Hz to 52.8 Hz, with a bandwidth of 3.4 Hz.

Figure 10a,b show the frequency responses of the harvester for the case without an IR and with a 2:1 IR at an acceleration of 0.6 g, respectively. By inducing multimodal interactions to generate two IRs, as shown in Figure 9a, the bandwidth of each subsystem increases by 267% compared to the case without an IR. This matches the improvement observed with a 2:1 internal resonance. Without an internal resonance, the total frequency bandwidth is the sum of the individual bandwidths of the two subsystems 2×0.6 Hz. However, when the 2:1:2 IR is tuned, the overall bandwidth increases up to 3.1 Hz. This represents a 2.6-fold improvement compared to using two separate subsystems without IRs. Although the 2:1:2 IR reduces the harvested power by 13% compared to the response without an IR, it represents a significant improvement over the 2:1 internal resonance, implying a reduction in power of 30%.

The Systematic Figure of Merit (${\mathrm{SFoM}}_{\mathrm{BW}}$) proposed by Liu et al. [50] was used to perform a comparative study of the performance between the designed harvester and state-of-the-art devices. The ${\mathrm{SFoM}}_{\mathrm{BW}}$ takes into account the average power density and the frequency bandwidth, normalized over the acceleration and size of the harvester, as follows:(19)$$\begin{array}{c} {\mathrm{SFoM}}_{\mathrm{BW}}=\frac{16\pi P_{\mathrm{av}}B}{X_{g}^{2}V_{\mathrm{tot}}} \end{array}$$
where $P_{\mathrm{av}}$ is the average power, B is the total frequency bandwidth, $X_{g}$ is the acceleration, and $V_{\mathrm{tot}}$ is the volume of the harvester.

Figure 11 compares the ${\mathrm{SFoM}}_{\mathrm{BW}}$ of the VEH device with various devices reported in the literature across a wide range of frequencies. The designed device demonstrates superior performance compared to existing vibration energy harvesters. The proposed VEH, which simultaneously activates two IRs, significantly enlarges the operational bandwidth. This performance can be further enhanced by optimizing the electromechanical coupling through adjustments to the coil characteristics, which could result in significantly better performance [51].

## 5. Conclusions

This work aims to enhance the bandwidth performance of a vibration energy harvester by simultaneously activating two IRs. An electromagnetic VEH device with a 2:1:2 IR is designed, consisting of three coupled oscillators in which nonlinear magnetic forces are used to tune the desired frequency ratio and introduce nonlinearity into the system. The numerical results predict two IRs with an M-shaped behavior, in which the frequency responses bend in opposite directions from the center frequency, resulting in a broader operational bandwidth. A multi-objective optimization procedure is performed to optimize the bandwidth and harvested power, in which adding mass perturbation leads to symmetrical behavior in the frequency response. A prototype VEH device is fabricated, and experimental investigations confirm the theoretical findings, demonstrating the simultaneous activation of two IRs. The results show that exploiting nonlinear multimodal interactions enhances the power density and significantly enlarges the bandwidth. At a base acceleration of 0.6 g, the bandwidth of the harvester is 2.6 times higher than that of a harvester without an IR. The proposed harvester demonstrates a significantly high figure of merit compared to previously reported devices in the literature, achieving an ${\mathrm{SFoM}}_{\mathrm{BW}}$ of 7600 kg/m^3^. This value highlights its high performance in maintaining high average power output across a wide range of frequencies. Parametric studies show that when the base excitation is increased from 0.4 g to 0.8 g, the bandwidth rises from 2.6 Hz to 3.4 Hz, which is an increase of approximately 31%. Future research will further develop this concept by exploring the generalization of multiple internal resonances in large-scale energy harvesters.

## Figures and Tables

**Figure 1 micromachines-16-00023-f001:**
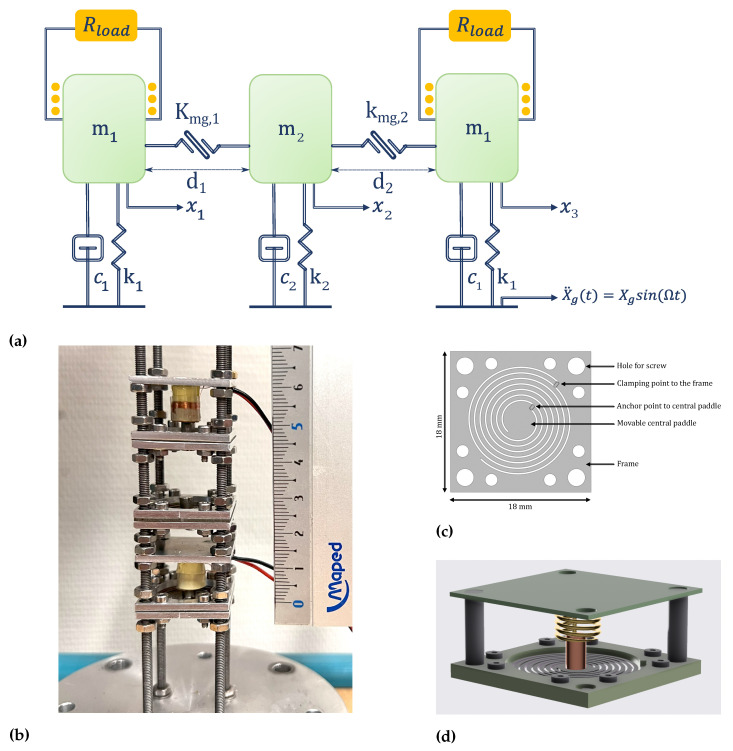
(**a**) Equivalent spring-mass model of the VEH device consisting of three coupled oscillators with a 2:1:2 IR, where $m_{1}$ and $m_{2}$ are the magnetic masses, and since $m_{3}=m_{1}$, $m_{3}$ is represented by $m_{1}$. $k_{1}$ and $k_{2}$ are the linear mechanical stiffnesses, and since $k_{3}=k_{1}$, $k_{3}$ is represented by $k_{1}$. $k_{mg,1}$ and $k_{mg,2}$ are the nonlinear coupling stiffnesses between the two repulsive magnets. $c_{1}$ and $c_{2}$ are the corresponding total damping coefficients. $x_{1}$, $x_{2}$, and $x_{3}$ are the relative displacements of the corresponding masses. (**b**) The proposed electromagnetic VEH device with three magnetically coupled oscillators. (**c**) The spiral-shaped spring. (**d**) Schematic of a single-subsystem electromagnetic VEH.

**Figure 2 micromachines-16-00023-f002:**
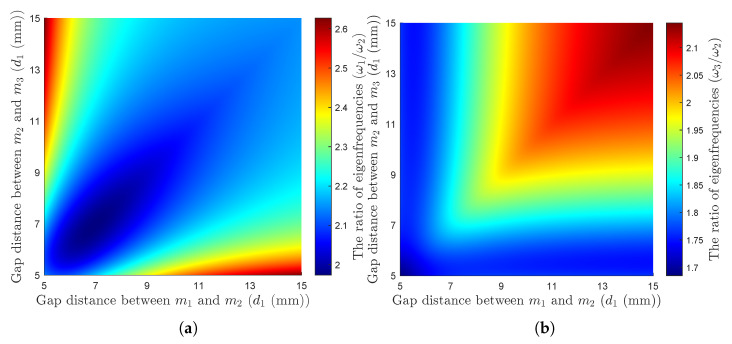
Variation in the ratio of the natural frequencies by adjusting the gap between the magnets $d_{1}$ and $d_{2}$: (**a**) the ratio of the natural frequencies $\omega_{1}/\omega_{2}$, (**b**) the ratio of the natural frequencies $\omega_{3}/\omega_{2}$.

**Figure 3 micromachines-16-00023-f003:**
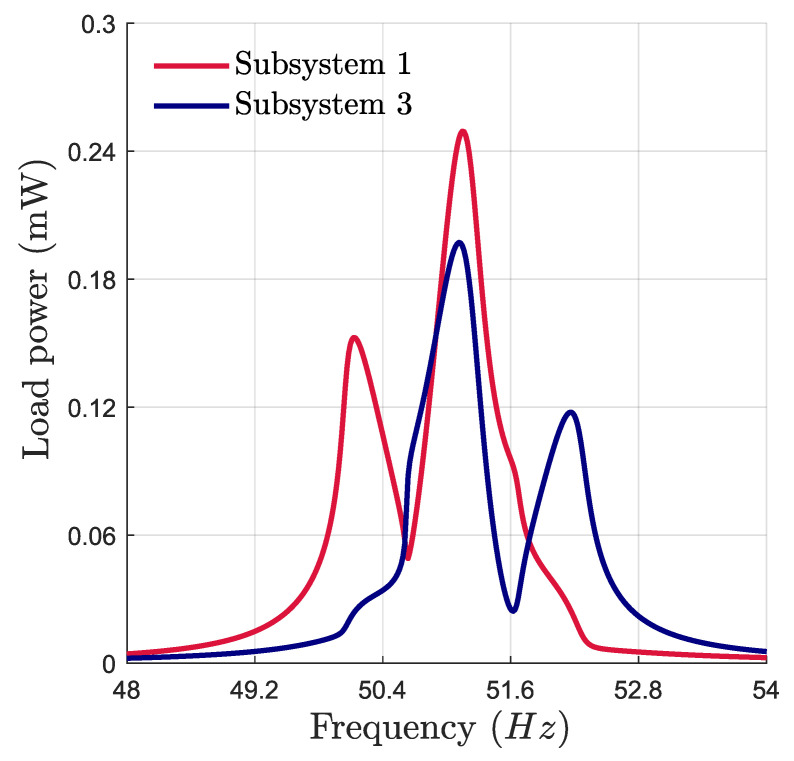
Numerical frequency response of the harvested power at an acceleration of 0.4 g.

**Figure 4 micromachines-16-00023-f004:**
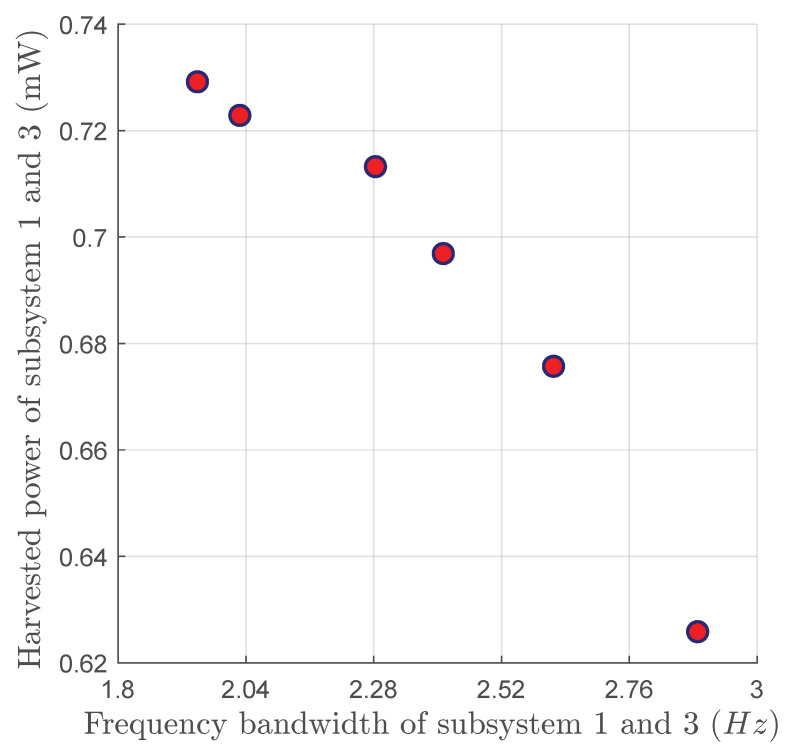
Pareto front of the frequency bandwidth and the harvested power for the VEH device with 2:1:2 IR at an acceleration of 0.4 g.

**Figure 5 micromachines-16-00023-f005:**
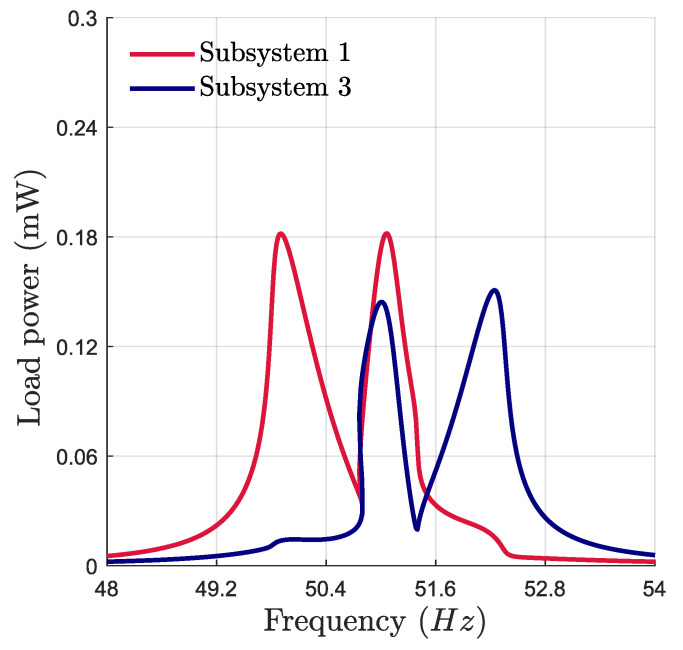
Numerical frequency response of the harvested power around the first primary resonance at an acceleration of 0.4 g.

**Figure 6 micromachines-16-00023-f006:**
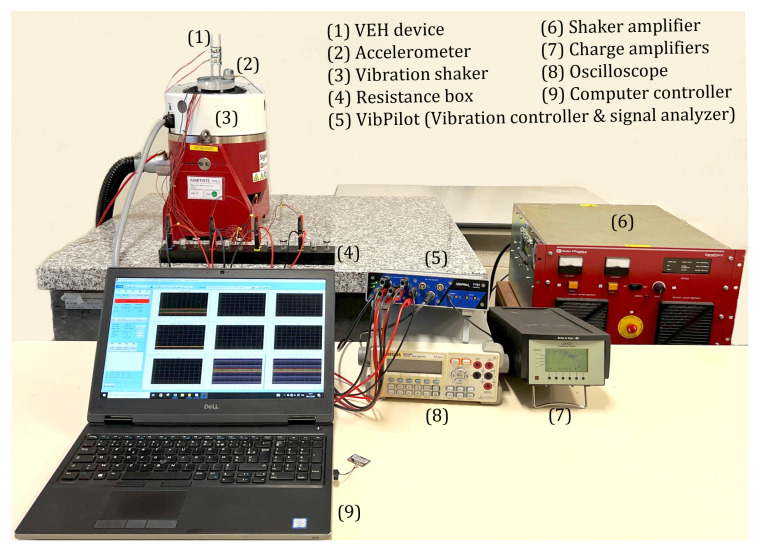
Vibration test setup for the dynamic characterization of the designed VEH.

**Figure 7 micromachines-16-00023-f007:**
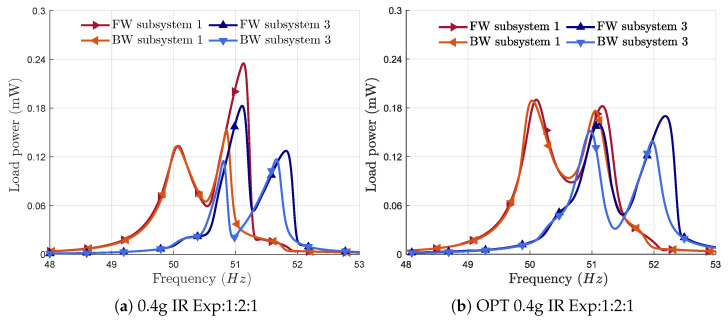
Frequency response curves of the harvested power around the first primary resonance at an acceleration of 0.4 g (**a**) before the optimization procedure (non-optimized results) and (**b**) after the optimization procedure (optimized results).

**Figure 8 micromachines-16-00023-f008:**
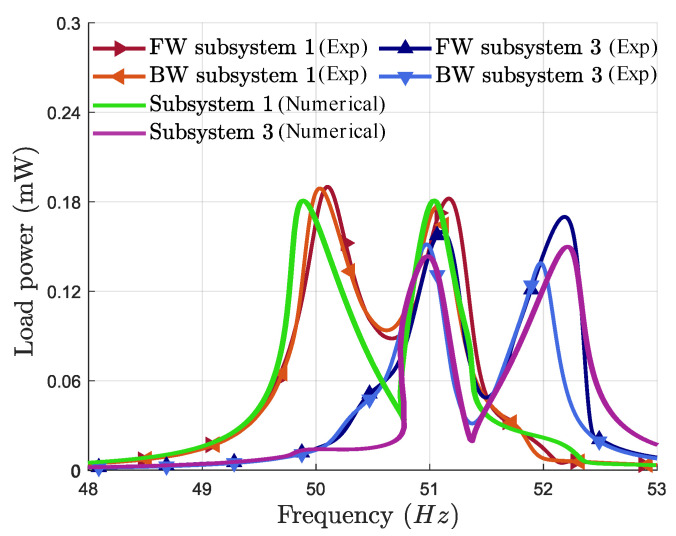
Comparison between the experimental and numerical frequency responses of the harvested power around the first primary resonance at an acceleration of 0.4 g (results obtained after optimization).

**Figure 9 micromachines-16-00023-f009:**
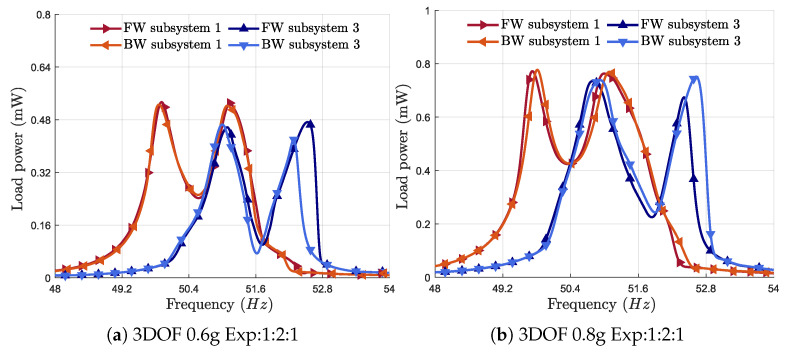
Frequency response of the harvested power around the first primary resonance, with optimized results at accelerations of (**a**) 0.6 g and (**b**) 0.8 g.

**Figure 10 micromachines-16-00023-f010:**
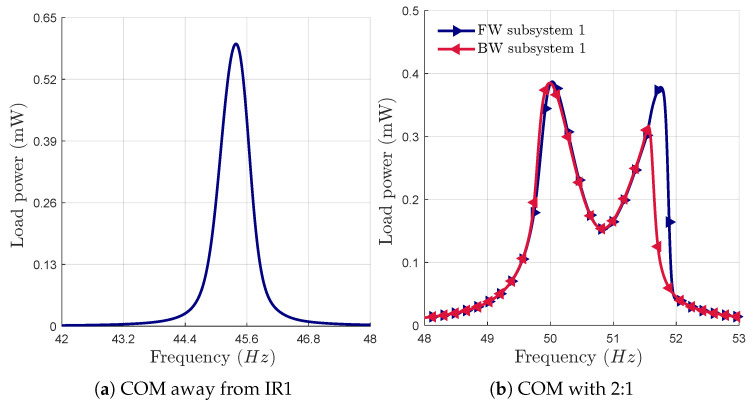
Frequency response curves of the harvested power at an acceleration of 0.6 g (**a**) without IR and (**b**) with a 2:1 IR.

**Figure 11 micromachines-16-00023-f011:**
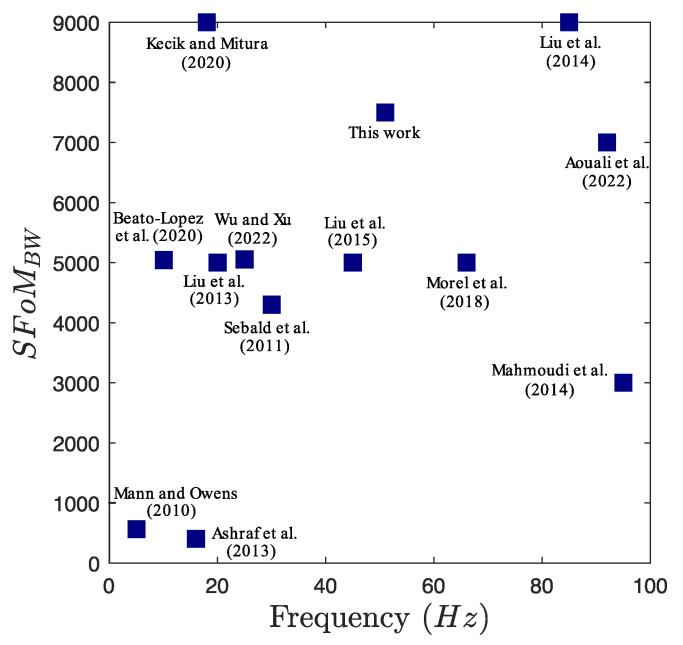
Comparison of the proposed harvester ${\mathrm{SFoM}}_{\mathrm{BW}}$ with current state-of-the-art devices [31,41,52,53,54,55,56,57,58,59,60,61].

**Table 1 micromachines-16-00023-t001:** Design parameters of the proposed electromagnetic VEH.

Parameter	Symbol	Value	Unit
Linear spring stiffness	$k_{1}$	105	N/m
	$k_{2}$	720	N/m
Mass	$m_{1}$	1.3	g
	$m_{2}$	1.8	g
Load resistance	$R_{load}$	25	Ω
Internal resistance of the coil	$R_{int}$	17	Ω
Mechanical damping coefficient	$\xi_{m}$	0.16%	−
Electromagnetic coupling coefficient	$\delta_{em}$	0.14	V.S/m

**Table 2 micromachines-16-00023-t002:** Optimal mass perturbations on subsystems 1 and 3.

Mass perturbation on subsystem 1 ($\alpha_{1}$)	1.01 (+1%)
Mass perturbation on subsystem 3 ($\alpha_{2}$)	0.955 (−4.5%)

**Table 3 micromachines-16-00023-t003:** Material and geometric parameters of the magnet and coil.

Component	Parameter	Value
Neodymium Magnet	MMagnetization	N45
	Residual magnetic field	1.37 (T)
	Height	6 (mm)
	Diameter	3 (mm)
	Magnetization moment	0.0465 (A $\cdot m^{2}$)
Coil	IInternal resistance	117 (Ω)
	Number of turns	73
	Diameter	5 (mm)

## Data Availability

The data that support the findings of this study are available within the article.

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
