# Peer review of "Enhancing the Performance of Vibration Energy Harvesting Based on 2:1:2 Internal Resonance in Magnetically Coupled Oscillators"

_micromachines, 2024, doi:10.3390/mi16010023_

Round 1
Reviewer 1 Report
Comments and Suggestions for Authors
The research topic is interesting.
The authors present a mathematical model, numerical results and an optimization process. Results are corroborated by experimental tests.
Main remark
The authors should improve the description of the system and the relation between the spring-mass model of Figure1a and the mathematical model of equations 4), 5) and 6).
In particular:
At line 101 it is stated that “a coil wraps each magnet”, but Figure 1a shows 2 coils.
An expression of the normalized damping coefficients should be given to highlight the contribution of electromagnetic damping.
Minor remarks
Line 203: “gamultiobj” please clarify that it is the name of MATLAB function.
Results in the frequency domain are presented in a very narrow frequency range. A plot with an extended range (e.g. 0-100 Hz) could help the reader understand the actual properties and potentialities of the proposed system.
An expression of power is not given in the mathematical model, but it is directly introduced in equation 14
Author Response
We are very much thankful to the reviewers for their deep and thorough review. We have revised our present research paper in the light of the reviewers useful suggestions and comments. We hope our revision has improved the paper to the level of the reviewer’s satisfaction and the required quality for the journal. Number wise answers to his specific comments/suggestions/queries are as follows. All the modifications made are in red in the revised version of the manuscript.
Reviewer 1
The research topic is interesting. The authors present a mathematical model, numerical results and an optimization process. Results are corroborated by experimental tests.
Main remark
The authors should improve the description of the system and the relation between the spring-mass model of Figure1a and the mathematical model of equations 4), 5) and 6).
In particular:
At line 101 it is stated that “a coil wraps each magnet”, but Figure 1a shows 2 coils.
An expression of the normalized damping coefficients should be given to highlight the contribution of electromagnetic damping.
Thank you for your valuable feedback. Further explanations are added to clarify the relationship between the spring-mass model and the mathematical model. Additionally, we appreciate your suggestion regarding the expression of the normalized damping coefficients. We include a more detailed explanation of the normalized damping coefficients to clearly highlight the contribution of electromagnetic damping.
Minor remarks
Line 203: “gamultiobj” please clarify that it is the name of MATLAB function.
Thank you for your comment. The sentence is revised to clarify that gamultiobj is the name of a MATLAB function, as follows:
"The MATLAB function gamultiobj, which implements a variation of the Non-Sorting Genetic Algorithm (NSGA-II), is used in this work."
Results in the frequency domain are presented in a very narrow frequency range. A plot with an extended range (e.g. 0-100 Hz) could help the reader understand the actual properties and potentialities of the proposed system.
Thank you very much for your insightful suggestion. The results are presented in a narrow frequency range to better highlight and clearly visualize the internal resonance phenomena, which are a key focus of this work. We believe that extending the frequency range may reduce the clarity and make it harder to observe these important details. For this reason, we have chosen to present the results in the current format. We hope this explanation addresses your concern, and we sincerely appreciate your understanding.
An expression of power is not given in the mathematical model, but it is directly introduced in equation 14
Thank you for pointing this out. We have revised the manuscript to ensure clarity. We have now included an explicit expression or derivation for power in the mathematical model section to provide a complete explanation before its introduction in Equation (14).
Reviewer 2 Report
Comments and Suggestions for Authors
This manuscript presents a novel approach to enhancing the performance of vibration energy harvesting systems through the use of a 2:1:2 internal resonance (IR) mechanism in magnetically coupled oscillators. The study proposes a device comprising three magnetically coupled oscillators with adjustable gap distances, enabling the tuning of eigenfrequencies to achieve 2:1:2 IR. The authors employ numerical simulations to predict the device's behaviour and implement a multi-objective optimization procedure to refine performance by introducing mass perturbations. It can be published as a journal paper by addressing the following questions
1. Why was the repulsive force configuration used in the model? Is it possible to achieve similar results using the attractive force configuration?
2. Figure 1b lacks clarity. It would be helpful to add more details or provide a view from a different ang le to improve understanding.
3. A reference is needed for Equation 3.
4. What is the magnetization moment in Equation 3? The paper does not provide a definition or explanation for this term.
5. Consider moving the equations for the coefficients of Equations 4, 5, and 6 out of the main body. These equations could be presented separately to enhance readability and avoid cluttering the main text.
6. Adding a graph to compare experimental results with numerical ones would be beneficial. This would strengthen the validation of the model and improve the overall impact of the results.
Author Response
We are very much thankful to the reviewers for their deep and thorough review. We have revised our present research paper in the light of the reviewers useful suggestions and comments. We hope our revision has improved the paper to the level of the reviewer’s satisfaction and the required quality for the journal. Number wise answers to his specific comments/suggestions/queries are as follows. All the modifications made are in red in the revised version of the manuscript.
Reviewer 2
This manuscript presents a novel approach to enhancing the performance of vibration energy harvesting systems through the use of a 2:1:2 internal resonance (IR) mechanism in magnetically coupled oscillators. The study proposes a device comprising three magnetically coupled oscillators with adjustable gap distances, enabling the tuning of eigenfrequencies to achieve 2:1:2 IR. The authors employ numerical simulations to predict the device's behaviour and implement a multi-objective optimization procedure to refine performance by introducing mass perturbations. It can be published as a journal paper by addressing the following questions.
- Why was the repulsive force configuration used in the model? Is it possible to achieve similar results using the attractive force configuration?
The repulsive force configuration is used because it stabilizes the system and ensures proper spacing between the harvesters. Attractive forces, on the other hand, add negative stiffness to the system, which reduces its ability to resist disturbances and may lead to dynamic instability. This instability can cause uncontrolled oscillations, misalignment, or even collisions between the harvesters, making it unsuitable for consistent and efficient energy harvesting.
- Figure 1b lacks clarity. It would be helpful to add more details or provide a view from a different ang le to improve understanding.
Thank you for your feedback. In response, a 3D schematic of a single subsystem (1 DOF) has been added in the revised manuscript to clarify the configuration and improve understanding.
- A reference is needed for Equation 3.
A reference is added for Equation (3) in the revised manuscript.
- What is the magnetization moment in Equation 3? The paper does not provide a definition or explanation for this term.
Thank you for your comment. The magnetization moment as well as the geometric parameters of the magnet and coil have been added in Table 3 in the revised manuscript.
- Consider moving the equations for the coefficients of Equations 4, 5, and 6 out of the main body. These equations could be presented separately to enhance readability and avoid cluttering the main text.
Thank you for your suggestion. The equations for the coefficients in Equations (4), (5), and (6) have been moved out of the main body and presented separately. This helps enhance readability and avoids cluttering the main text, making it easier for the reader to follow the derivations.
- Adding a graph to compare experimental results with numerical ones would be beneficial. This would strengthen the validation of the model and improve the overall impact of the results.
Thank you for your suggestion. To address this, we added in the revised manuscript Figure 8 and its description to illustrate the good agreement between the experimental results and the numerical ones.
Reviewer 3 Report
Comments and Suggestions for Authors
This paper focuses on an electromagnetic vibration energy harvester with 2:1:2 internal resonance (IR), aiming to enhance performance in bandwidth and harvested power. The research is significant as it explores a novel approach in the field of vibration energy harvesting. However, some aspects need further clarification and improvement. Here are the comments:
1. Regarding the multi-objective optimization process, the use of mass perturbations to optimize bandwidth and power is a valid approach. However, it is not clear how the chosen mass perturbation values (α1 = 1.01 (+1%) and α2 = 0.955 (-4.5%)) were determined to be the optimal ones. Were there any other candidate solutions that were considered and why were they rejected? A more detailed analysis of the optimization process, such as showing the convergence behavior of the algorithm and the trade-off between different solutions on the Pareto front, would enhance the credibility of the results.
2. When comparing the performance of the proposed harvester with other devices in the literature using the Systematic Figure of Merit (SFoMBW), it is essential to ensure that the comparison is fair and comprehensive. Were all the compared devices tested under the same or similar conditions? Are there any other important performance metrics that could be considered for a more in-depth comparison? For example, the efficiency of energy conversion or the durability of the device over time could also be relevant factors.
3. Table 1 provides a useful summary of the design parameters. However, it would be helpful to add a brief description of each parameter in the table caption or in the text near the table. This would assist readers in quickly grasping the significance of each value without having to search through the text for explanations.
4. In the experimental section, the description of the test setup is quite detailed, which is good. However, when presenting the experimental results, it would be useful to include information about the repeatability of the measurements. For example, stating the number of trials conducted for each experiment and the standard deviation of the measured values (such as power and bandwidth) would give readers a better sense of the reliability of the data. In Figure 7, 8, and 9, the power-frequency response curves are shown. It would be beneficial to add legends that clearly distinguish between the different subsystems and the optimized/non-optimized cases. This would make it easier for readers to quickly identify and compare the different curves.
Author Response
We are very much thankful to the reviewers for their deep and thorough review. We have revised our present research paper in the light of the reviewers useful suggestions and comments. We hope our revision has improved the paper to the level of the reviewer’s satisfaction and the required quality for the journal. Number wise answers to his specific comments/suggestions/queries are as follows. All the modifications made are in red in the revised version of the manuscript.
REVIEWER 3:
This paper focuses on an electromagnetic vibration energy harvester with 2:1:2 internal resonance (IR), aiming to enhance performance in bandwidth and harvested power. The research is significant as it explores a novel approach in the field of vibration energy harvesting. However, some aspects need further clarification and improvement. Here are the comments:
- Regarding the multi-objective optimization process, the use of mass perturbations to optimize bandwidth and power is a valid approach. However, it is not clear how the chosen mass perturbation values (α1 = 1.01 (+1%) and α2 = 0.955 (-4.5%)) were determined to be the optimal ones. Were there any other candidate solutions that were considered and why were they rejected? A more detailed analysis of the optimization process, such as showing the convergence behavior of the algorithm and the trade-off between different solutions on the Pareto front, would enhance the credibility of the results.
Thank you for your insightful comment. The mass perturbation values (α1 = 1.01 (+1%) and α2 = 0.955 (-4.5%)) were determined through a targeted parametric study aimed at enhancing both power and bandwidth. These values were specifically chosen to achieve the desired ratio between eigenfrequencies, ensuring the occurrence of two internal resonances. In the Pareto front presented in Figure 4, the selected solution achieves the maximum frequency bandwidth while maintaining conditions favorable for maximizing the harvested power.
- When comparing the performance of the proposed harvester with other devices in the literature using the Systematic Figure of Merit (SFoMBW), it is essential to ensure that the comparison is fair and comprehensive. Were all the compared devices tested under the same or similar conditions? Are there any other important performance metrics that could be considered for a more in-depth comparison? For example, the efficiency of energy conversion or the durability of the device over time could also be relevant factors.
Thank you for your comment. We ensured a fair comparison by selecting devices from the literature tested under similar conditions, such as comparable acceleration levels and frequency ranges. We recognize the importance of other performance metrics, such as energy conversion efficiency and long-term durability. While our current comparison focuses on SFoMBW, we plan to include these additional factors in future work to provide a more comprehensive evaluation of the harvester's performance.
- Table 1 provides a useful summary of the design parameters. However, it would be helpful to add a brief description of each parameter in the table caption or in the text near the table. This would assist readers in quickly grasping the significance of each value without having to search through the text for explanations.
Thank you for your valuable feedback. In response to your suggestion, we have provided a more detailed explanation in the relevant sections of the manuscript.
- In the experimental section, the description of the test setup is quite detailed, which is good. However, when presenting the experimental results, it would be useful to include information about the repeatability of the measurements. For example, stating the number of trials conducted for each experiment and the standard deviation of the measured values (such as power and bandwidth) would give readers a better sense of the reliability of the data. In Figure 7, 8, and 9, the power-frequency response curves are shown. It would be beneficial to add legends that clearly distinguish between the different subsystems and the optimized/non-optimized cases. This would make it easier for readers to quickly identify and compare the different curves.
Thank you for your thoughtful suggestions. We would like to confirm that the experimental results demonstrate repeatability throughout the validation process, ensuring the reliability of the measurements. Each experiment was conducted 10 times to verify consistency and minimize any potential variability in the results. As for Figures 7, and 8, we have made modifications to the captions to enhance clarity considering the optimized and non-optimized results.